# Polysemous Language Gaussian Splatting via Matching-based Mask Lifting

## Abstract

Lifting 2D open-vocabulary understanding into 3D Gaussian Splatting (3DGS) scenes is a critical challenge. However, mainstream methods suffer from three key flaws: (i) their reliance on costly per-scene retraining prevents plug-and-play application; (ii) their restrictive monosemous design fails to represent complex, multi-concept semantics; and (iii) their vulnerability to cross-view semantic inconsistencies corrupts the final semantic representation. To overcome these limitations, we introduce MUSplat, a training-free framework that abandons feature optimization entirely. Leveraging a pre-trained 2D segmentation model, our pipeline generates and lifts multi-granularity 2D masks into 3D, where we estimate a foreground probability for each Gaussian point to form initial object groups. We then optimize the ambiguous boundaries of these initial groups using semantic entropy and geometric opacity. Subsequently, by interpreting the object's appearance across its most representative viewpoints, a Vision-Language Model (VLM) distills robust textual features that reconciles visual inconsistencies, enabling open-vocabulary querying via semantic matching. By eliminating the costly per-scene training process, MUSplat reduces scene adaptation time from hours to mere minutes. On benchmark tasks for open-vocabulary 3D object selection and semantic segmentation, MUSplat outperforms established training-based frameworks while simultaneously addressing their monosemous limitations.

## 1 Introduction

Open-vocabulary 3D scene understanding enables the parsing of 3D scenes with arbitrary natural language queries, moving beyond the limitations of predefined categories to offer enhanced generalization and richer semantics for applications like autonomous driving and robotics. The primary challenge in this domain lies in finding an efficient and effective 3D scene representation. Traditional methods such as voxels, point clouds, and meshes, while useful for structure modeling, struggle with the trade-off between detail and computational expense. Recently, 3D Gaussian Splatting (3DGS) (Kerbl et al., 2023) has provided a compelling solution by merging the explicit structure of traditional methods with the efficiency of neural techniques. It achieves high-quality modeling and rendering while maintaining high rendering speeds, making it an ideal foundation for next-generation 3D scene understanding.

More recently, several methods have leveraged 3DGS for point-level open-vocabulary 3D scene understanding, achieving remarkable results. Current research is largely dominated by the **training**-based contrastive learning paradigm (Li et al., 2025; Wu et al., 2024), as illustrated in Fig. 1(a). These methods rely on a laborious optimization pipeline: they first perform tens of thousands of mask-guided contrastive learning iterations to embed semantic features into each Gaussian, followed by clustering and post-processing steps to achieve feature-language alignment. Furthermore, a minority of studies (Jun-Seong et al., 2025) have explored a feature projection paradigm, which trains a specialized model to compress Contrastive Language-Image Pre-training (CLIP) (Radford et al., 2021) features and subsequently assigns these compressed features to the 3D Gaussians for open-vocabulary understanding. While these approaches have shown promising results, we argue that they suffer from three critical limitations that hinder their performance and practical deployment: **1) Reliance on Expensive Optimization:** The dominant contrastive learning paradigm requires tens of thousands of iterations for per-scene semantic optimization. Similarly, the feature

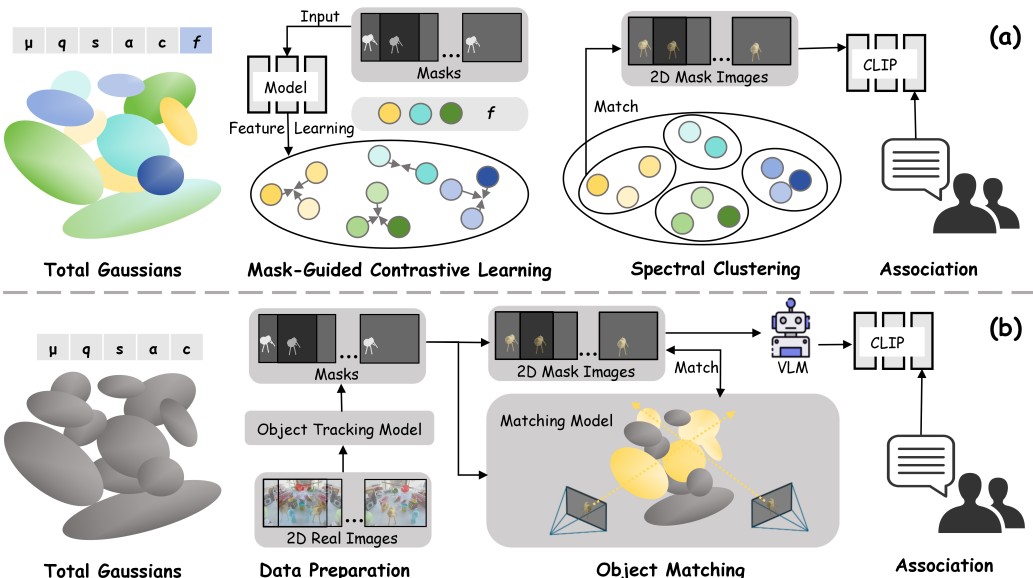

Figure 1: Pipelines for open-vocabulary understanding of 3D Gaussian scenes. (a) Training-based approaches: semantic features are learned via mask-guided contrastive learning, followed by clustering and post-processing. (b) Our training-free matching-based approach: semantic associations are determined directly through matching, without feature learning.

projection paradigm necessitates pre-training a specialized feature compression model. Both pathways introduce significant time and computational overhead, undermining the efficiency advantages of 3DGS. **2) Deficient Semantic Representation:** Existing methods lack richness and precision in their semantic representations. On one hand, the prevailing contrastive methods are restricted by a "one-to-one" semantic assignment, attributing a single semantic concept to each Gaussian. This makes them incapable of capturing polysemy, the phenomenon where a single point may belong to multiple semantic concepts (e.g., a desk point being simultaneously "desk", "wooden", and "furniture"). On the other hand, feature projection methods suffer from degraded feature quality. Their reliance on lossy feature compression often impairs the model's fine-grained understanding and semantic discriminability. **3) Fragile Multi-view Feature Aggregation:** A critical yet often overlooked issue is the viewpoint variance of CLIP's image features, where the same object instance yields significantly inconsistent embeddings across different viewpoints. Current aggregation strategies fail to robustly handle view-dependent variations. For instance, mainstream methods typically assign single-view masked CLIP features to Gaussians, while projection-based approaches perform a weighted average of multi-view features; both tactics lead to inaccurate 3D semantic representations. These concerns motivate the central question of our work: *"Can we construct a framework for open-vocabulary understanding of 3D Gaussian scenes that robustly aggregates multi-view features, supports polysemous representations, and operates without requiring any additional training?"*

To answer this question, we introduce **M**atching-based **U**nderstanding with 3D Gaussian **Splat**ting (**MUSplat**), a novel training-free framework that bypasses feature optimization entirely. As illustrated in Fig. 1(b), our framework determines the semantics of each Gaussian through a **matching** mechanism, which is inherently designed to handle polysemy and operate without per-scene retraining. Our framework first lifts 2D masks to form 3D object groups, then refines their boundaries with neutral point processing for enhanced precision. After that, We leverage a Vision-Language Model (VLM) to generate textual features for each object, enabling precise language-based retrieval. Our contributions are summarized as follows: **1)** We introduce MUSplat, a training-free and polysemy-aware framework for open-vocabulary understanding in 3D Gaussian scenes. **2)** We propose an object-level grouping method that achieves precise 3D instance grouping by probabilistically lifting 2D masks and subsequently refining ambiguous boundaries with a neutral point processing mechanism. **3)** We present a VLM-based distillation technique that forges a robust and unified textual representation from inconsistent multi-view visual features.

## 2 RELATED WORKS

### 2.1 PRELIMINARY: 3D GAUSSIAN SPLATTING

3D Gaussian Splatting (3DGS) (Kerbl et al., 2023) models 3D scenes with explicit 3D Gaussians, enabling high-quality, real-time rendering. It represents a scene as a collection of 3D Gaussians $\mathcal{G} = \{g_i\}_{i=1}^N$, each defined by its position, covariance (governing scale and orientation), color, and opacity. To generate a 2D image, these 3D Gaussians are projected onto an image plane and then blended in a depth-sorted order via "splatting". The final color $C(p)$ for any pixel $p$ is determined through alpha compositing (Munkberg et al., 2022):

$$C(p) = \sum_{i=1}^{|\mathcal{G}_p|} c_{g_i^p} \alpha_{g_i^p} \prod_{j=1}^{i-1} (1 - \alpha_{g_j^p}), \tag{1}$$

where $c_{g_i^p}$ and $\alpha_{g_i^p}$ are the color and opacity of the $i$-th Gaussian in the sorted set for pixel $p$. The product term, $\prod_{j=1}^{i-1}(1 - \alpha_{g_j^p})$, calculates the accumulated transmittance, which represents the light that reaches the $i$-th Gaussian after passing through all prior ones.

### 2.2 OPEN VOCABULARY UNDERSTANDING BASED ON 3DGS

Prevailing methods for semanticizing 3DGS for open-vocabulary understanding follow two primary paradigms: pixel-based and point-based. Pixel-based methods employ a "render-then-match" paradigm: they first render the entire scene into dense 2D feature maps and subsequently perform semantic matching in the image space. In contrast, point-based methods adopt a "match-then-render" strategy, first identifying a sparse set of semantically relevant 3D points and then rendering only this pre-filtered subset.

In pixel-based methods, Feature-3DGS (Zhou et al., 2024) distills semantic features from 2D foundation models into 3DGS, enabling fast semantic rendering. LEGaussians (Shi et al., 2024) adds uncertainty and semantic to each Gaussian and compares rendered semantic maps with quantized CLIP and DINO features. LangSplat (Qin et al., 2024) learns language features in a scene-specific latent space and renders them as semantic maps. GS-Grouping (Ye et al., 2024) assigns a compact identity encoding to each Gaussian and leverages masks from the Segment Anything Model (SAM) (Kirillov et al., 2023) for supervision. GOI (Qu et al., 2024) introduces an optimizable semantic hyperplane to separate pixels relevant to language queries, improving open-vocabulary accuracy.

However, these methods rely on rendered 2D semantic maps, so reasoning remains in 2D. They lack awareness of 3D structure and are thus less suited for tasks requiring direct 3D interaction, such as embodied intelligence. To overcome these limitations, point-based methods have been proposed. These approaches operate on a foundational "select-then-render" pipeline. OpenGaussian (Wu et al., 2024) uses SAM masks to learn instance features with 3D consistency, introduces a two-stage codebook for feature discretization, and links 3D points with 2D masks and CLIP features for open-vocabulary selection. InstanceGaussian (Li et al., 2025), based on Scaffold-GS (Lu et al., 2024), jointly learns appearance and semantics and adaptively aggregates instances, reducing semantic–appearance misalignment. Dr.Splat (Jun-Seong et al., 2025) trains a feature compressor for each scene and assigns compressed CLIP features to every Gaussian. All these approaches rely on computationally expensive iterative training. Our model instead follows a training-free matching framework that establishes a semantic link between 3D Gaussian points and a query by directly matching their uncompressed CLIP features.

## 3 METHOD

We propose a training-free pipeline for point-level 3D open-vocabulary semantic segmentation, as shown in Fig. 2. Our method takes a pre-trained 3DGS scene representation and its corresponding image sequence as input. First, the Data Preparation stage (§3.1) generates multi-view, multi-granularity object masks. Next, the Object-level Grouping stage (§3.2) links 2D masks to 3D Gaussian points and purifies the resulting object boundaries by identifying and excluding ambiguous neutral points. Finally, the Instance Feature Extraction stage (§3.3) uses a VLM to extract textual features for each object, enabling alignment with open-vocabulary queries.

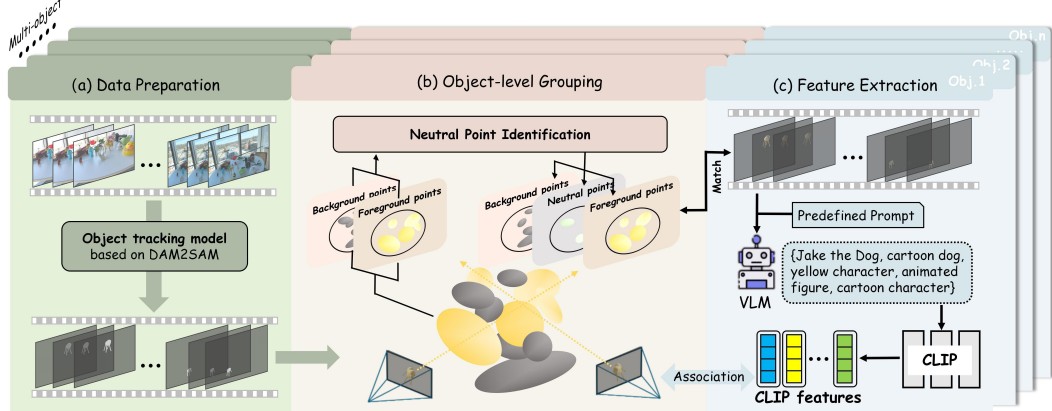

Figure 2: Overview of our method. (a) Multi-view 2D segmentation masks are first extracted from the input scene. (b) Based on these masks, our method lifts the objects into 3D point groups via back-projection, refining their boundaries by filtering ambiguous neutral points. (c) Each refined group is then grounded by using a VLM to generate textual hypotheses from its key views, which are encoded into semantic features via a CLIP text encoder.

## 3.1 DATA PREPARATION

Our goal is to extract a comprehensive set of multi-view segmentation masks for all objects in a scene. To this end, we first employ SAM on the initial frame, $I_0$, leveraging its ability to produce object masks at three distinct granularity levels (e.g., part, object, scene). To ensure stable tracking throughout the sequence, especially in complex scenes with visually similar distractors, we employ the DAM2SAM (Videnovic et al., 2025) model. Its specialized distractor-aware memory is crucial for maintaining accurate object identities where other trackers might fail. To capture new objects that appear later, we introduce a periodic detection mechanism that re-segments the scene at fixed intervals and identifies new instances based on a minimal IoU overlap criterion with existing tracks. The entire pipeline, from tracking to new object detection, is executed independently for each of the three granularity levels to yield a complete and hierarchical set of masks.

Our pipeline is designed for robustness, as potential data preparation artifacts, such as tracking failures or re-identification errors, are gracefully handled by our downstream object grouping and query matching modules. This design minimizes the requirements for perfect input data (see Appendix A.1 for details).

## 3.2 OBJECT-LEVEL GROUPING

To precisely group 3D Gaussian instances, we resolve ambiguous boundary points that corrupt segmentation using a two-stage, coarse-to-fine strategy. We first identify the object's high-confidence core via mask back-projection, then refine its boundaries by identifying and excluding ambiguous points with our neutral point processing module, ensuring a clean result.

**Initial 3D Grouping via Mask Back-projection.** We link 2D masks to 3D Gaussian points by back-projecting them to estimate a per-point foreground probability. For each object, we process its multi-view masks, first discarding any null (entirely black) masks from viewpoints where the object is unseen. For each valid mask, we then cast a ray through each pixel $r$ and sum the contributions of all intersected Gaussians. The contribution of the $j$-th Gaussian $G_j$ along ray $r$ is determined by its accumulated transmittance and opacity, defined as:

$$w(r, G_j) = T(r, G_j) \cdot \alpha(r, G_j), \tag{2}$$

where $T(r, G_j)$ denotes the accumulated transmittance up to $G_j$, and $\alpha(r, G_j)$ is its effective opacity. To ensure design consistency, we define the weight $w(r, G_j)$, representing the contribution of Gaussian $G_j$ to pixel $r$, to be identical to the forward color rendering weight of 3DGS given in Eq. 1.

For each 3D Gaussian point $G_j$, we compute its total foreground ($W_1$) and background ($W_0$) weights by aggregating contributions from multi-view 2D masks:

$$W_k(G_j) = \sum_{v \in \mathcal{V}} \sum_{r \in \mathcal{P}_v} \delta(m_v(r) - k) \cdot w_v(r, G_j), \quad k \in \{0, 1\}, \tag{3}$$

where $\mathcal{V}$ is the set of visible views, $\mathcal{P}_v$ the pixels in a view, $m_v(r)$ the mask value, $\delta(\cdot)$ the indicator function, and $w_v(r, G_j)$ the contribution weight. Based on these weights, we form an initial set of foreground points, $\mathcal{F}$, using a simple hard assignment: $\mathcal{F} = \{G_j \mid W_1(G_j) > W_0(G_j)\}$. All remaining points are consequently assigned to the background.

**Neutral Point Processing.** During rendering, it is inevitable that some points lie at the boundaries between objects but do not semantically belong to any specific category. We refer to these points as neutral points. Their semantic assignment directly affects the accuracy of rendered object boundaries. Existing methods typically assume that each 3D Gaussian point belongs either to the foreground or to the background, i.e., every point has a clear semantic label. In practice, however, many points at boundaries are transitional and may not carry a well-defined semantic meaning. Such points should be considered neither foreground nor background. Our goal is to identify and exclude these neutral points from semantic supervision, thereby mitigating potential artifacts and improving the accuracy of the final segmentation.

To identify neutral points, we leverage multi-view semantic consistency. While points deep within an object are consistently labeled across views, those near boundaries often exhibit conflicting semantics. To quantify this ambiguity, we treat each viewpoint as providing a discrete semantic label for a given Gaussian point. Specifically, for each point $p$, we project its center into every visible view and record whether it lands inside (foreground) or outside (background) the corresponding 2D mask. This process yields a set of binary labels $\{l_v\}_{v \in \mathcal{V}}$ for each 3D point. The semantic entropy $H(p)$, which quantifies the disagreement among these discrete labels, is calculated as:

$$H(p) = -\left( \frac{V_f}{V} \log_2 \frac{V_f}{V} + \frac{V_b}{V} \log_2 \frac{V_b}{V} \right), \tag{4}$$

where $V_f$ and $V_b$ are the respective counts of foreground and background labels within the set $\{l_v\}_{v \in \mathcal{V}}$, and $V = |\mathcal{V}| = V_f + V_b$. Points with entropy $H(p)$ exceeding a threshold $\tau_h$ form an initial candidate set $\mathcal{C}$ of ambiguous points.

This set $\mathcal{C}$ is impure, containing both true neutral points used for smooth blending and mislabeled solid points that belong to an object's surface. To distinguish them, we use a geometric property: opacity ($\alpha$). Points on solid surfaces typically have high opacity, while transitional points used for anti-aliasing have low opacity. We filter $\mathcal{C}$ based on this idea: if a point $p \in \mathcal{C}$ has an opacity $\alpha(p) > \tau_\alpha$, we classify it as a mislabeled solid point. These points, identified as part of a solid surface, are removed from the neutral candidate set $\mathcal{C}$, thereby retaining their initial classification as either foreground or background.

The remaining points in $\mathcal{C}$ are confirmed as the final neutral point set, which is excluded from all semantic supervision. The final set of foreground points is thus defined by the expression $\mathcal{F} \setminus \mathcal{C}$. Likewise, the background set is refined by removing these same points. We use fixed values for the thresholds $\tau_h$ and $\tau_\alpha$ across all experiments for simplicity and robustness. A detailed sensitivity analysis on their selection is provided in Appendix B.2.

### 3.3 INSTANCE FEATURE EXTRACTION

To enable open-vocabulary understanding, we extract semantic features for each 3D Gaussian cluster. As illustrated in Fig. 3(a), prevailing methods approach this by extracting visual features from multi-view masks of all objects. The semantic feature for a given 3D cluster is then derived by either selecting the feature from the most similar mask or by computing a weighted average of features from all associated masks. However, this approach suffers from two major drawbacks. First, it overlooks the inherent semantic inconsistency across viewpoints. The visual features of the same object instance can vary significantly from different views (see Appendix E.3 for a visualization), leading to a biased and inaccurate 3D semantic representation. Second, extracting and storing CLIP features for every object mask across all views incurs substantial computational and storage overhead, rendering the process inefficient. To avoid this, we propose semantic distillation: we use

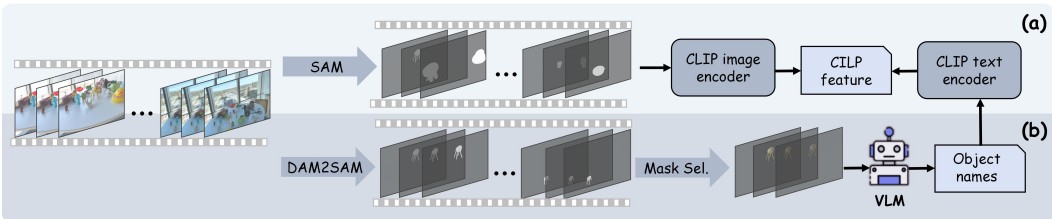

Figure 3: Comparison of 2D-3D Feature Association Pipelines. (a) Mainstream Method (via direct extraction): All object masks, typically generated by SAM, are used to directly extract CLIP image features. (b) Our Method (via semantic distillation): We leverage DAM2SAM to track a single instance. The top-N most visible masks are then interpreted by a VLM, distilling volatile visual appearances into a stable CLIP text representation derived from the generated object identity.

a VLM to interpret key views and generate textual hypotheses of the object's identity. As shown in Fig. 3(b), this converts volatile visual appearances into a stable, canonical text representation, providing a robust foundation for open-vocabulary matching.

Specifically, for each object, we select the top-$N$ masked views with the largest visible areas and feed them into a VLM along with a predefined text prompt, which instructs the model to generate a set of candidate names describing the object (see Appendix A.1 for details). Our framework is VLM-agnostic; for this study, we employ Gemini 2.5 Pro (Comanici et al., 2025) , but other models can be readily substituted (see Appendix B.2 for an ablation on VLM choice). Using a pre-trained CLIP text encoder, we encode the candidate names into a feature set $\mathbf{Q}$ and an open-vocabulary text query into a feature vector $\mathbf{s}$. We quantify their semantic relevance using cosine similarity:

$$\text{sim}(\mathbf{s}, \mathbf{q}) = \frac{\mathbf{s} \cdot \mathbf{q}}{\|\mathbf{s}\|\|\mathbf{q}\|}, \tag{5}$$

We then form a set of matching features, $\mathbf{Q}_m$, by selecting all candidates whose similarity to the query exceeds a threshold $\eta$: $\mathbf{Q}_m = \{\mathbf{q} \in \mathbf{Q} \mid \text{sim}(\mathbf{s}, \mathbf{q}) > \eta\}$. The final segmentation for the query is the union of all 3D Gaussian point sets associated with the features in $\mathbf{Q}_m$.

## 4 Experiments

### 4.1 Open-Vocabulary Object Selection in 3D Space

**Settings 1) Task.** Given a text query as input, the task is to produce multi-view renderings of the semantically corresponding 3D instance(s). First, the textual feature of the input query is extracted using the CLIP model. Then, cosine similarity is computed between the query feature and the textual features of each instance, and the most similar instance(s) are selected. Finally, all 3D Gaussian points belonging to the selected instances are rendered into multi-view images through the 3DGS rasterization pipeline. **2) Baselines.** We compare our method with several recent representative approaches, including Dr.Splat (Jun-Seong et al., 2025), OpenGaussian (Wu et al., 2024), LangSplat (Qin et al., 2024), LEGaussian (Shi et al., 2024), InstanceGaussian (Li et al., 2025), Feature-3DGS (Zhou et al., 2024), GS-Grouping (Ye et al., 2024), and GOI (Qu et al., 2024). These approaches fall into the two primary categories of point-based and pixel-based methods. To provide a clear comparison, we detail the comparative aspects such as training time and search thresholds for these methods in Tab. 1. **3) Dataset.** We adopt the LERF (Kerr et al., 2023) dataset, annotated by LangSplat. This dataset consists of multi-view images capturing 3D scenes with long-tail objects and provides ground-truth 2D annotations for texture-level queries. For a fair comparison, we use the same predefined query texts as those used in OpenGaussian.

**Results 1) Quantitative Evaluation.** As shown in Tab. 2, our method achieves a new state-of-the-art (SOTA) result, outperforming the previous best-performing method by 10.7 mIoU. Although our method does not operate at the pixel level, its performance on average surpasses that of SOTA pixel-based approaches. As shown in Tab. 1, our zero-shot framework eliminates per-scene optimization and training, reducing feature storage by nearly 1000x and significantly lowering VRAM overhead. These results demonstrate that our approach delivers SOTA performance without the substantial computational overhead of previous methods. **2) Qualitative Evaluation.** Qualitative results are

Table 1: This caption compares computational resources for the LERF `figurines` scene, including per-scene optimization time, peak VRAM use during object-level grouping, and storage for instance feature extraction. Our method is highly efficient, cutting CLIP feature storage from gigabytes to megabytes and using the least amount of VRAM. Notably, it is the only method that works directly in 3D without any training. Note that "–" marks methods that do not use CLIP features.

| Method | Venue | Domain | Scene Opt. | Train Time | CLIP Feat. Stor. | Peak VRAM |
|---|---|---|---|---|---|---|
| LEGaussians | CVPR'24 | 2D | Required | ∼2h | ∼3GB | ∼20 GB |
| LangSplat | CVPR'24 | 2D | Required | ∼2h | ∼3GB | ∼20 GB |
| Feature-3DGS | CVPR'24 | 2D | Required | ∼1h | ∼3GB | ∼26 GB |
| GS-Grouping | ECCV'24 | 2D | Required | ∼1h | – | ∼28 GB |
| GOI | MM'24 | 2D | Required | ∼1h | – | ∼24 GB |
| OpenGaussian | NIPS'25 | **3D** | Required | ∼1h | ∼3GB | ∼22 GB |
| InstanceGaussian | CVPR'25 | **3D** | Required | ∼2h | ∼3GB | ∼24 GB |
| Dr.Splat | CVPR'25 | **3D** | **None** | ∼1h | ∼3GB | ∼24 GB |
| Ours | – | **3D** | **None** | **None** | **∼3MB** | **∼8 GB** |

Table 2: mIoU results for open-vocabulary object selection in 3D space on the LERF dataset. **Bold**/Underline indicates the best/second-best performance per category.

| Method | Venue | ramen | teatime | figurines | Waldo_kitchen | Mean |
|---|---|---|---|---|---|---|
| **Pixel-based** | | | | | | |
| LEGaussians | CVPR'24 | 46.0 | 60.3 | 40.8 | 39.4 | 46.6 |
| LangSplat | CVPR'24 | 51.2 | **65.1** | **44.7** | **44.5** | **51.4** |
| Feature-3DGS | CVPR'24 | 43.7 | 58.8 | 40.5 | 39.6 | 45.7 |
| GS-Grouping | ECCV'24 | 45.5 | 60.9 | 40.0 | 38.7 | 46.3 |
| GOI | MM'24 | **52.6** | 63.7 | 44.5 | 41.4 | 50.6 |
| **Point-based** | | | | | | |
| LangSplat-m | CVPR'24 | 6.1 | 16.6 | 8.3 | 8.3 | 9.8 |
| LEGaussians-m | CVPR'24 | 15.8 | 19.3 | 18.0 | 11.8 | 16.2 |
| OpenGaussian | NIPS'25 | 31.0 | 60.4 | 39.3 | 22.7 | 38.4 |
| InstanceGaussian | CVPR'25 | 24.6 | 63.4 | 45.5 | 29.2 | 40.7 |
| Dr.Splat(Top-40) | CVPR'25 | 24.7 | 57.2 | 53.4 | 39.1 | 43.6 |
| **Ours** | – | **45.6** | **64.4** | **66.4** | **40.9** | **54.3** |

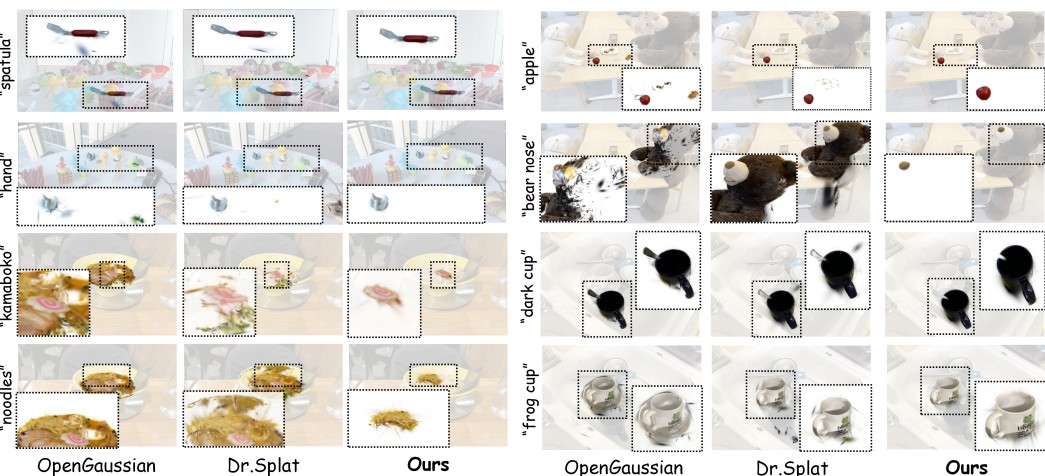

Figure 4: Qualitative results on object selection from the LERF dataset. OpenGaussian fails to separate nearby objects or maintain sharp boundaries, while InstanceGaussian struggles to capture fine-grained details. In contrast, our method correctly interprets fine-grained instructions to generate precise selections with well-defined boundaries.

Table 3: Quantitative results for open-vocabulary 3D semantic segmentation on the ScanNet dataset.

| Method | Venue | 19 classes | | 15 classes | | 10 classes | |
|---|---|---|---|---|---|---|---|
| | | mIoU↑ | mAcc↑ | mIoU↑ | mAcc↑ | mIoU↑ | mAcc↑ |
| LangSplat-m | CVPR'24 | 3.8 | 9.1 | 5.4 | 13.2 | 8.4 | 22.1 |
| LEGaussians-m | CVPR'24 | 1.6 | 7.9 | 4.6 | 16.1 | 7.7 | 24.9 |
| OpenGaussian | NIPS'25 | 24.7 | 41.5 | 30.1 | 48.3 | 38.3 | 55.2 |
| InstanceGaussian | CVPR'25 | 40.7 | 54.0 | 42.5 | 59.1 | 47.9 | 64.0 |
| Dr.Splat(Top-40) | CVPR'25 | 29.6 | 47.7 | 38.2 | 60.4 | 50.8 | 73.5 |
| **Ours** | – | **45.5** | **58.4** | **47.2** | **61.7** | **53.7** | **74.9** |

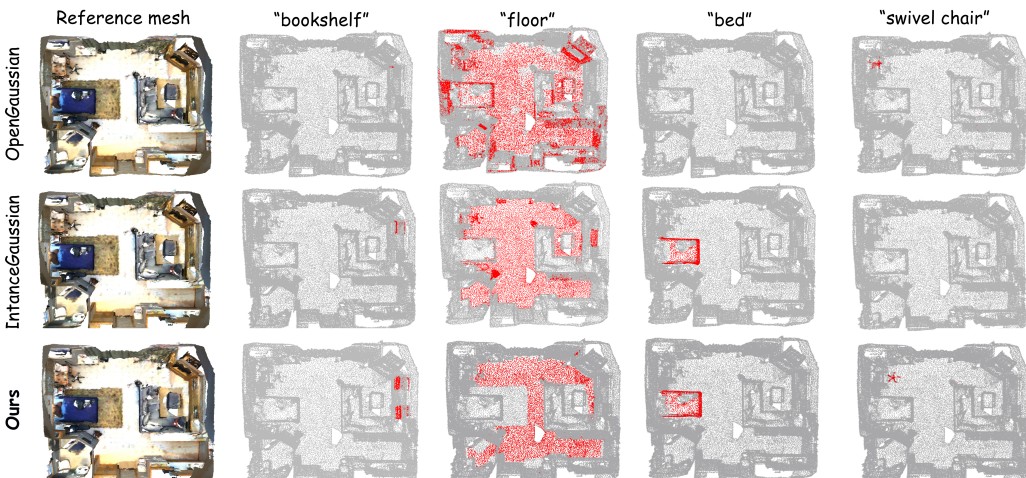

Figure 5: Qualitative results of our 3D object segmentation on the ScanNet dataset. OpenGaussian and InstanceGaussian rely on matching CLIP features extracted from 2D images. This approach is susceptible to feature inconsistencies arising from different mask viewpoints, often leading to incorrect matches (e.g., for the bed and chair). In contrast, our method achieves accurate 3D segmentation with sharp and well-defined boundaries.

shown in Fig. 4. Relying on spatial clustering, OpenGaussian often yields incorrect matches with imprecise boundaries and includes irrelevant points (e.g., spatula, apple). Dr.Splat struggles with fine-grained instructions (e.g., bear nose, noodles), as its compressed CLIP features fail to capture full semantic richness, leading to interpretation errors. In contrast, our method uses uncompressed CLIP representations for linguistic precision, while our independent assignment and neutral-point handling mechanisms ensure sharp boundaries and superior performance.

## 4.2 OPEN-VOCABULARY 3D SEMANTIC SEGMENTATION

**Settings 1) Task.** The objective is to automatically extract 3D Gaussian points corresponding to input class names (e.g., wall, chair, table). The segmented Gaussian points are then converted into a point cloud to be evaluated against the ground-truth annotated point cloud. To ensure a precise correspondence between the converted point cloud and the ground truth, we disable the 3D Gaussian densification process during training. **2) Baselines.** Consistent with the object selection task, we compare our method against several recently proposed approaches: Dr.Splat (Jun-Seong et al., 2025), InstanceGaussian (Li et al., 2025), OpenGaussian (Wu et al., 2024), LangSplat-m, and LEGaussians-m. LangSplat-m and LEGaussians-m are adaptations of existing pixel-based methods (Shi et al., 2024; Qin et al., 2024), specifically modified to perform direct 3D referring operations. As this task requires a direct understanding of 3D points, pixel-based methods are not applicable and are therefore excluded from our comparison. **3) Dataset.** We employ the Scan-Net (Dai et al., 2017) dataset, a large-scale benchmark comprising indoor scene data with calibrated RGB-D images and 3D point clouds annotated with ground-truth semantic labels. For a fair and direct comparison, we adopt the same scenes and evaluation categories used in OpenGaussian.

**Results 1) Quantitative Analysis.** Tab.3 shows the performance on the ScanNet dataset using text queries for 19, 15, and 10 of its classes. The results show that our method consistently achieves SOTA segmentation performance across all scenes relative to the baselines. Notably, our method is training-free, which highlights its efficiency and precision. **2) Qualitative Analysis.** Qualitative results are presented in Fig.5. In complex scenes from ScanNet, both OpenGaussian and Instance-Gaussian frequently exhibit incorrect matches, which limits their accuracy. This limitation arises from their reliance on matching masked CLIP image features, as semantic inconsistencies across different viewing angles make it difficult for such methods to achieve high-precision results. In contrast, by leveraging a VLM to acquire instance-level features, our method demonstrates correct segmentation with sharp and clear boundaries.

## 4.3 ABLATION STUDY

**Neutral Point Processing.** We ablate our neutral point processing on the LERF dataset with results in Tab. 4. Case #1 is the baseline without any filtering. Case #2 adds our entropy-based filtering. Case #3 introduces our full model, which further incorporates an opacity filter. As shown, entropy filtering alone provides a minor gain by suppressing noise, but its aggressive nature can inadvertently remove valid foreground points. Adding the opacity filter resolves this issue, achieving the highest mIoU. This demonstrates that both stages are complementary and essential for the final performance.

**Instance Feature Extraction.** To demonstrate the advantages of using a VLM for language feature extraction, we compare our approach with three baselines derived from the CLIP image encoder. Case #1 uses the feature from the single view with the largest mask area. Case #2 averages features from all valid views. Case #3 first renders the class foreground points onto each view and computes the IoU between the rendered foreground masks and candidate masks. We then discard the low-IoU masks and average the features of the remaining ones. The results are presented in Tab. 5. Single-view methods struggle to capture comprehensive semantics, while multi-view averaging methods often yield ambiguous features due to occlusions. Although filtering-based methods significantly improve matching accuracy, they require a rendering pass for each view, which incurs high runtime costs. Moreover, these methods can obscure discriminative details due to feature discrepancies across different views (see Appendix E.3 for details). In contrast, our VLM-based method distills these multi-view cues into a consistent textual representation, effectively capturing the nuanced attributes required for complex and abstract queries.

Table 4: Ablation on neutral point processing.

| Case | Entropy Fil. | Opacity Fil. | mIoU↑ |
|------|------------|-------------|-------|
| #1   |            |             | 53.0  |
| #2   | ✓          |             | 53.2  |
| #3   | ✓          | ✓           | **54.3** |

Table 5: Ablation on feature extraction.

| Case | Method | mIoU↑ |
|------|--------|-------|
| #1   | Single-View Image Mat. | 36.9 |
| #2   | Averaged Image Mat. | 39.2 |
| #3   | Filtered Image Mat. | 50.1 |
| #4   | VLM-Text Mat. (Ours) | **54.3** |

## 5 CONCLUSION AND LIMITATION

In this work, we introduced MUSplat, a training-free and plug-and-play model for open-vocabulary understanding of 3D Gaussian scenes. Our approach shifts the focus from feature learning to direct matching, supported by a back-projection mechanism for initial grouping, a neutral point process for boundary refinement, and a VLM that distills visual appearance into a robust textual representation. Evaluations on the LERF dataset and other benchmarks confirm that MUSplat delivers SOTA-level performance on point-level open-vocabulary tasks at a fraction of the computational cost. These results suggest that matching-based solutions can serve as viable alternatives to current training-based paradigms for point-level open-vocabulary understanding in 3DGS.

Despite its strong performance, our method has certain limitations: 1) The accuracy of our object-level grouping can be compromised in instances of substantial inaccuracies in the initial segmentation masks from SAM. 2) On rare occasions, the VLM may assign incorrect semantic labels to objects within the masks. For a detailed analysis of failure cases, please refer to Appendix D. Addressing these edge cases offers promising avenues for future research.

**Ethics Statement**

We affirm our commitment to the ICLR Code of Ethics in all aspects of this research. The work presented is purely methodological, utilizing standard, publicly available benchmarks, with no involvement of human subjects. We have carefully considered the potential for negative societal impacts and find no foreseeable issues concerning fairness, bias, privacy, or security directly related to our proposed methods. The research is intended to advance the field for beneficial and constructive applications.

**Reproducibility Statement**

To ensure full reproducibility, we provide comprehensive technical details, hyperparameters, and experimental results in the main paper and appendix. All experiments were conducted on publicly available datasets. We commit to releasing our source code upon publication to facilitate verification and future research.

**The Use of Large Language Models**

In accordance with ICLR 2026 guidelines, we disclose that Google's Gemini 2.5 Pro was utilized as a writing assistant to polish and improve the clarity of this manuscript. The LLM's role was strictly confined to refining author-written drafts for grammar and style. All scientific content, including claims, methodologies, and results, originated solely from the human authors, who critically reviewed every suggestion and assume full responsibility for the accuracy and originality of this work.

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

APPENDIX

# A  IMPLEMENTATION DETAILS

## A.1  MODEL IMPLEMENTATION DETAILS

**Data Preparation.** Initially, we employ SAM with grid-based point prompting to acquire initial static object masks at varying granularities from the first input frame, $I_0$. Subsequently, these masks extracted from $I_0$ are utilized by the DAM2SAM (Videnovic et al., 2025) model to track the corresponding objects throughout the entire image sequence.

To ensure all objects appearing throughout the sequence are captured, we introduce a periodic new-object detection mechanism. This check is performed at a fixed interval of $\Delta t = 10$ frames. At each check, we first compute the total area of all tracked masks in the current frame, $A_t$. We then trigger a full re-segmentation on this frame using SAM to get a candidate mask area, $A_{cand}$. A potential new object event is flagged if the ratio $A_t/A_{cand}$ falls below a threshold $\tau_{area} = 0.9$. When triggered, we identify a mask from the candidate set as a "new" object if its maximum Intersection over Union (IoU) with any existing tracked mask is below a threshold of $\tau_{iou} = 0.6$. Once identified, these new objects are added to the tracking pool and propagated by DAM2SAM henceforth.

Existing research (Lu et al., 2025) suggests that such a detection mechanism can introduce two potential drawbacks: (1) tracking failures for some objects, resulting in incomplete object tracks, and (2) re-appearing objects being misidentified as new after their tracking has been lost, leading to a single object being assigned multiple instance IDs. Our model, however, does not need to overcome these issues during the data preparation stage.

Regarding the first issue, we simply discard views with empty masks (i.e., where object tracking has failed) during our object-level grouping stage. As demonstrated in Appendix B.2, our model achieves robust performance even with a reduced number of views per object. Consequently, this issue has a negligible impact on the overall model accuracy.

Regarding the second issue, the emergence of multiple instances for a single object is handled by our matching process. The matching between open-vocabulary queries and instance point clusters is a one-to-many operation based on similarity. In the event of multiple matches, we take the union of their results as the final output. Therefore, the presence of multiple instances for the same object does not degrade the final matching accuracy.

In summary, our model imposes minimal requirements on the data preparation stage and functions effectively even with partial mask information for each object. This demonstrates the robustness of our approach to imperfections in the input data.

**Object-Level Grouping.** The object-level grouping process is accomplished within a single forward rendering pass. In our implementation, we simply accumulate the contribution weights of all participating 3D Gaussians during the forward pass of the 3D Gaussian Splatting render. Throughout this process, the contribution weight of each Gaussian is naturally aggregated, obviating the need for auxiliary data structures or redundant computations. By leveraging the highly optimized volumetric projection inherent to 3D Gaussian Splatting, our method achieves exceptional computational efficiency while maintaining semantic coherence. For the subsequent neutral point processing, we use fixed thresholds across all experiments to ensure robustness and consistency. The semantic entropy threshold is set to $\tau_h = 0.9$, and the opacity threshold for filtering is set to $\tau_\alpha = 0.1$. A detailed sensitivity analysis for these hyperparameters is provided in Appendix B.2.

**Instance Feature Extraction.** We acquire features for each object instance as follows. First, we identify the three largest masks for the instance based on pixel area. For each selected mask, we highlight the corresponding object on the original image with a green bounding box, creating three distinct input images. These images are then processed by a VLM, which generates a set of five nouns that describe the instance.

To match an instance against a user's text query, we compute the cosine similarity between the CLIP feature embedding of the query and the CLIP embeddings of the five nouns associated with that instance. This design allows a single query to potentially match multiple instances. A match is

deemed successful if the similarity score for *any* of an instance's five candidate nouns surpasses a predefined threshold of $\eta = 0.9$.

The specific prompt template used to elicit these nouns from the VLM is defined as follows:

```
In the images, identify the object that is
enclosed by a bright green outline.  Provide
five distinct and appropriate nouns to describe
ONLY that specific object.  Return ONLY
the five nouns separated by slashes (e.g.,
car/automobile/vehicle/motorcar/transport).  Do not
add any other explanatory text, titles, or formatting.
```

### A.2 LERF DATASET EVALUATION

We evaluate our model on the LERF dataset, using annotations from LangSplat. Due to the absence of 3D ground truth, we follow the 2D-based evaluation protocol from OpenGaussian. This protocol measures 3D understanding by computing the multi-view IoU accuracy between rendered occupancy masks from our selected 3D Gaussians and the ground-truth masks, which were manually annotated and provided by OpenGaussian for a set of text queries.

### A.3 SCANNET DATASET EVALUATION

For evaluation on the ScanNet dataset, we select the same 10 scenes as used in OpenGaussian: scene0000_00, scene0062_00, scene0070_00, scene0097_00, scene0140_00, scene0200_00, scene0347_00, scene0400_00, scene0590_00, and scene0645_00.

The 19 categories defined by ScanNet used for text queries are: wall, floor, cabinet, bed, chair, sofa, table, door, window, bookshelf, picture, counter, desk, curtain, refrigerator, shower curtain, toilet, sink, and bathtub. 15 categories are without picture, refrigerator, shower curtain, bathtub; 10 categories are further without cabinet, counter, desk, curtain, sink.

## B ADDITIONAL EXPERIMENTAL RESULTS

### B.1 ADDITIONAL QUALITATIVE RESULTS

Fig. 6 presents additional qualitative results for the task of object selection in 3D space on the LERF dataset. Fig. 7 showcases more results of our model on the open-vocabulary 3D semantic segmentation task on the ScanNet dataset. These results were not included in the main manuscript due to space limitations. Consistent with our previous observations, both OpenGaussian and InstanceGaussian exhibit limitations in handling object boundaries and in fine-grained semantic understanding. In contrast, our model yields results with significantly sharper and more accurate semantic interpretations.

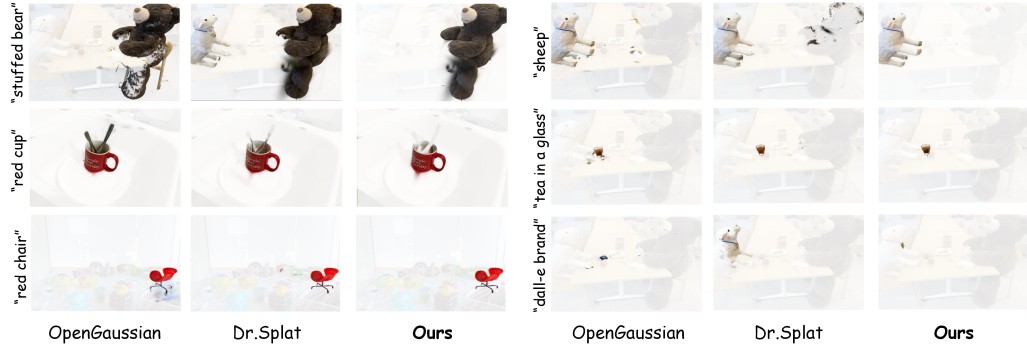

Figure 6: Additional qualitative results for open-vocabulary object selection on the LERF dataset.

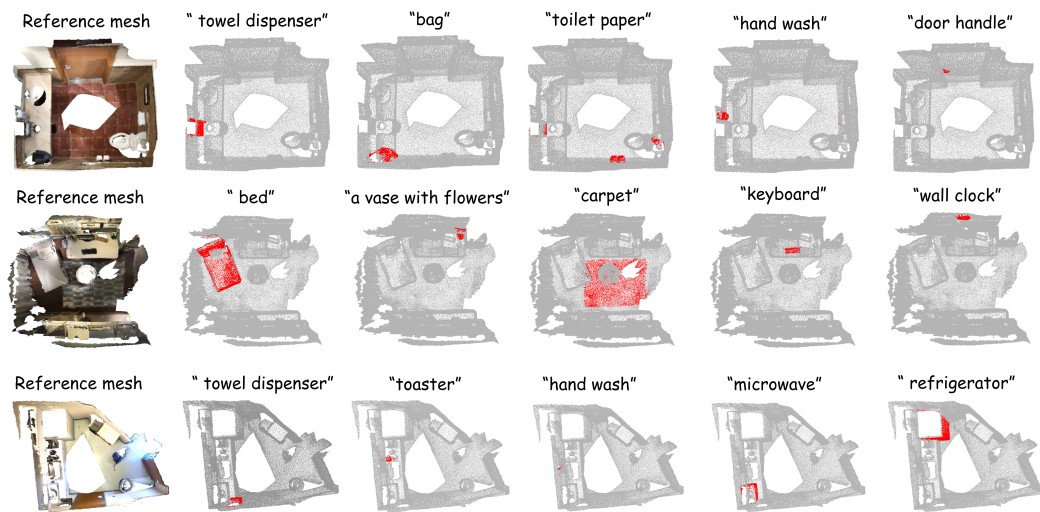

Figure 7: Additional qualitative results for open-vocabulary 3D semantic segmentation on the Scan-Net dataset.

## B.2 ADDITIONAL ABLATION STUDIES

**Scene Understanding with Limited Mask Supervision.** Our method leverages a mask-matching mechanism for semantic understanding, a characteristic that enables it to perform 3D segmentation from only a sparse set of 2D masks. To validate this capability, we conduct experiments using progressively sparser subsets of 2D masks (corresponding to $1/2$, $1/4$, $1/8$, $1/16$, and $1/32$ of the total available views), while all other model settings are held constant. Finally, we perform an open-vocabulary 3D object extraction task and qualitatively evaluate the results.

As illustrated in the Fig. 8, our method exhibits high robustness to the number of provided masks. Even with masks from only $1/8$ of the views, our method maintains high-quality segmentation. This demonstrates our model's high data efficiency and its ability to generalize from sparse supervision. However, when the number of masks becomes excessively sparse, such as at $1/16$ or $1/32$, a portion of the 3D Gaussians may not be observed by any masked camera view. This lack of supervision results in noticeable artifacts. Notably, the $1/32$ subset often corresponds to merely 5–10 foreground masks. While these extreme cases produce artifacts, the ability to generate a coherent result from such minimal data underscores our method's low reliance on dense supervision and corroborates its strong generalization capabilities.

**Ablation Study on Neutral Point Thresholds.** On the LERF dataset, we investigate the influence of the entropy threshold $\tau_h$ and the opacity threshold $\tau_\alpha$ in our two-stage neutral point processing module. The results of this sensitivity analysis are presented in Tab. 8. The baseline configuration, which bypasses entropy-based filtering by setting $\tau_h = 1.0$, achieves an mIoU of 53.0. A notable improvement is observed when $\tau_h$ is lowered to 0.9, underscoring the efficacy of pruning points with high semantic ambiguity.

The necessity of the subsequent opacity-based filtering is also validated. With $\tau_h = 0.9$, setting $\tau_\alpha = 0$ removes all high-entropy points indiscriminately and degrades performance to 53.2 mIoU. This suggests that high-entropy points with high opacity are geometrically significant and should be retained. Peak performance is achieved at $(\tau_h, \tau_\alpha) = (0.9, 0.1)$, which obtains an mIoU of 54.3. This configuration strikes a favorable trade-off between removing ambiguous transitional points and preserving geometrically salient structures. While the model demonstrates reasonable robustness to other settings, further reductions in $\tau_h$ to 0.8 or 0.5 yield diminished returns, likely due to the erroneous exclusion of valid surface points. Based on these findings, we adopt $\tau_h = 0.9$ and $\tau_\alpha = 0.1$ for all main experiments.

**Instance Feature Extraction.** The core of our instance feature extraction module is a VLM that grounds textual queries to 3D visual features. The representational capacity of the VLM is therefore a critical determinant of performance. To investigate this dependency, we ablate the VLM compo-

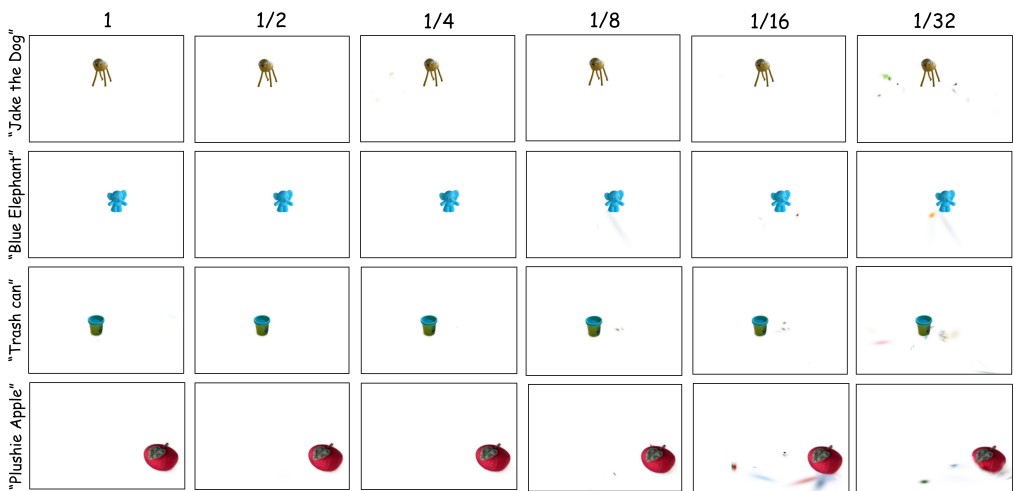

Figure 8: Open-Vocabulary 3D Object Extraction from Sparse Masks. We perform an open-vocabulary 3D object extraction task on the `figurines` scene from the LERF dataset, providing a progressively smaller subset of 2D masks as supervision. The results demonstrate that our model's accuracy experiences negligible degradation when using $\geq 1/4$ of the total masks. With only $1/8$ of the masks, it still exhibits a strong capability to capture the object's geometry. Even in the extreme case with as few as $1/32$ of the masks, our model can still recover the object's coarse shape.

nent with three different pre-trained models on the LERF dataset: SenseNova 6.5 Pro, InternVL3-78B (Chen et al., 2024), and Gemini 2.5 Pro (Comanici et al., 2025) .

The results, presented in Tab. 6, reveal a strong positive correlation between the representational power of the VLM and final segmentation accuracy. More specifically, employing VLMs known for more robust vision-language grounding consistently yields substantial gains in mIoU. This indicates that the quality of the semantic features provided by the VLM is a critical determinant of performance in this task. Therefore, the performance ceiling of our model is not static; it is set to rise in tandem with the ongoing evolution of Vision-Language Models.

We further analyze the method's sensitivity to the number of descriptive text prompts used for instance matching on the LERF dataset. As shown in Tab. 7, the relationship between prompt quantity and segmentation accuracy is non-monotonic. Starting from a single prompt, performance improves as the number of descriptors increases to five. This suggests that a richer set of semantic cues helps the VLM disambiguate instances, particularly for concepts too nuanced to be captured by a single term. However, increasing to 10 prompts leads to performance degradation. We hypothesize that an excessive number of prompts may introduce semantic noise or redundant information, thereby interfering with the VLM's feature matching process. Consequently, we use five descriptive prompts, as this configuration strikes a favorable balance between semantic richness and feature ambiguity.

### B.3 OPEN-VOCABULARY 3D OBJECT EDITING

Our method enables open-vocabulary editing of objects in 3DGS scenes by mapping a language query to corresponding instance IDs and then applying targeted manipulations. Fig. 9 demonstrates the scene editing capabilities of our method. Starting from an original scene reconstructed via 3DGS, we can select an object to perform operations such as **removal** (Fig 9(a)), **translation** (Fig. 9(b)), or **stylization** (Fig. 9(c)).

### B.4 OPEN-VOCABULARY OBJECT EXTRACTION IN COMPLEX AND REAL-WORLD SCENES

To evaluate our model's comprehension capabilities in complex scenes, we conduct experiments on the Grasp-Net dataset (Fang et al., 2023). This dataset is characterized by challenging object arrangements, including overlapping, adjacent, and contained instances. Despite the close proximity between instances, our model successfully distinguishes and segments them. As shown in Fig. 10,

Table 6: Ablation on the choice of VLM.

| Model | mIoU↑ |
|---|---|
| SenseNova 6.5 Pro | 47.0 |
| InternVL3-78B | 50.2 |
| Gemini 2.5 Pro | **54.3** |

Table 7: Ablation on number of prompts.

| Number of Prompts | mIoU↑ |
|---|---|
| 1 | 44.0 |
| 3 | 50.9 |
| 5 | **54.3** |
| 10 | 53.6 |

Table 8: Ablation on neutral point processing thresholds $\tau_h$ and $\tau_\alpha$.

| $\tau_h$ | $\tau_\alpha$ | mIoU↑ |
|---|---|---|
| 1.00 | / | 53.0 |
| 0.99 | 0.1 | 53.8 |
| 0.90 | 0.5 | 53.8 |
| 0.90 | 0.1 | **54.3** |
| 0.90 | 0.01 | 54.2 |
| 0.90 | 0.0 | 53.2 |
| 0.80 | 0.1 | 53.8 |
| 0.50 | 0.1 | 53.1 |

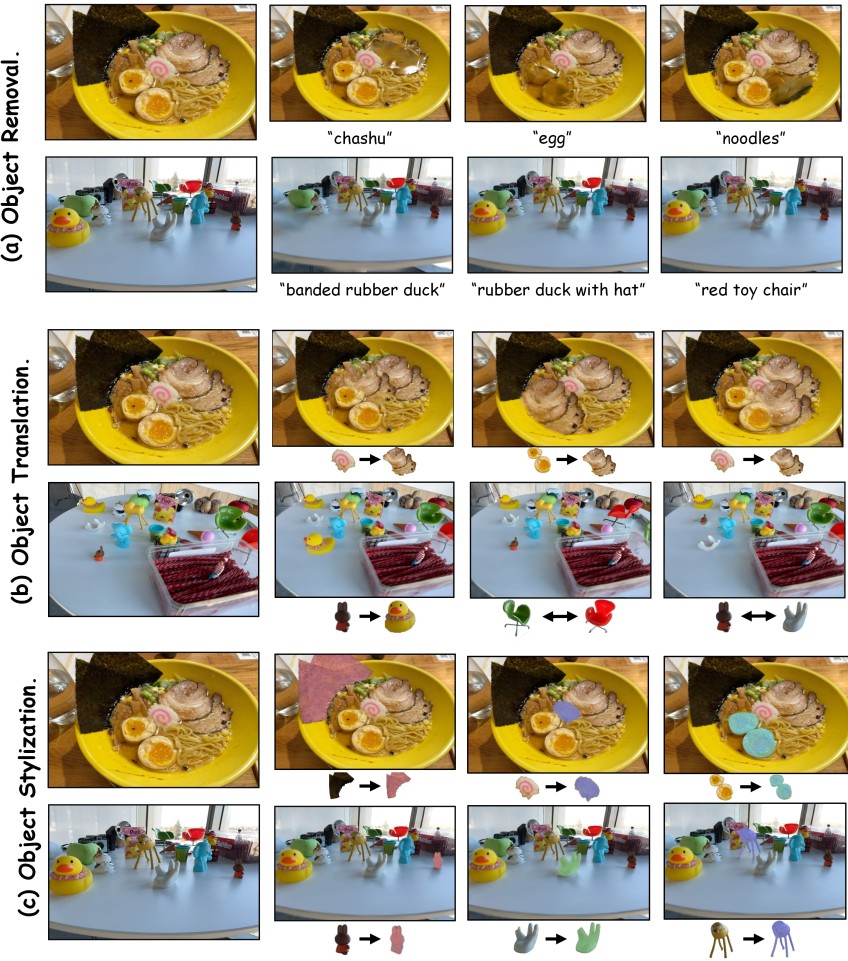

Figure 9: Demonstration of our scene editing capabilities. (a) Object Removal. (b) Object Translation. (c) Object Stylization. All manipulations are applied directly to the 3D scene rather than on the 2D rendered images.

our method produces sharp, well-defined rendering boundaries, demonstrating its effectiveness in such challenging scenarios.

Furthermore, to assess its practical applicability, we validate our method on a real-world scene. We captured an office environment using a standard mobile phone and tasked our model with open-vocabulary object extraction. The qualitative results, presented in Fig. 11, demonstrate that our model performs robustly on this in-the-wild data. This highlights the method's strong generalization capabilities and its potential for real-world applications.

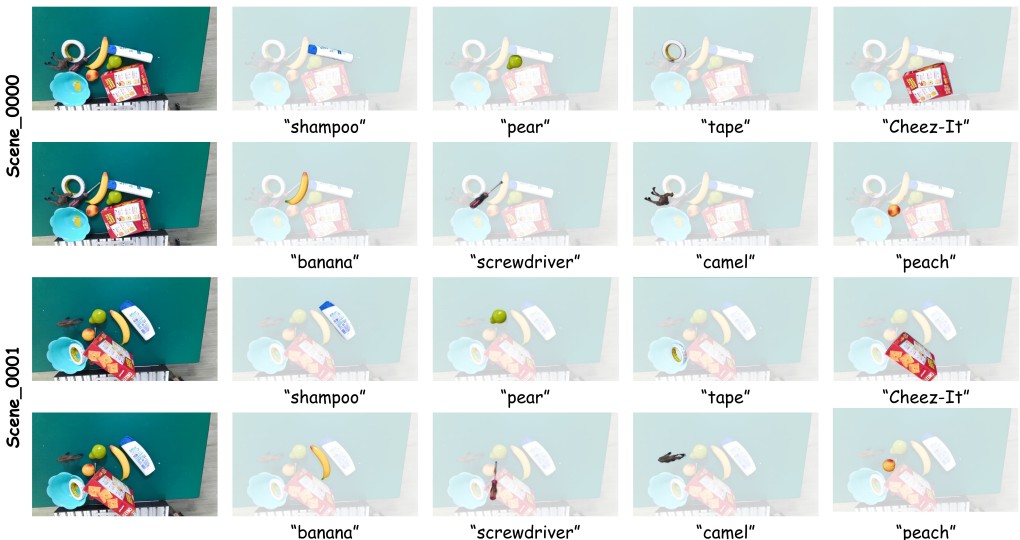

Figure 10: Qualitative results for the open-vocabulary object extraction task on the Grasp-Net dataset.

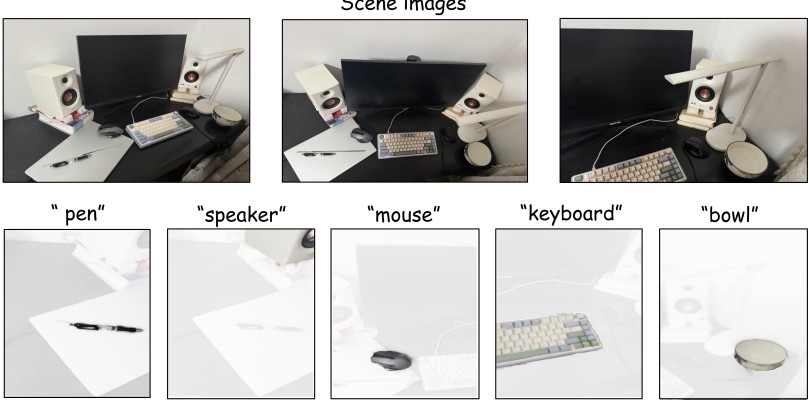

Figure 11: Qualitative results for the open-vocabulary object extraction task on a real-world scene captured with a mobile phone.

## C EFFICIENCY ANALYSIS

To dissect our method's efficiency, we provide a detailed component-wise runtime breakdown in Tab. 9, based on the `teatime` scene in the LERF dataset, which contains 131 distinct instance categories. The total end-to-end processing time for this complex scene is approximately 9.25 minutes (555.14s), including all computational and I/O stages. The results clearly identify the primary

computational bottlenecks, with three stages accounting for over 99% of the total computational workload: VLM Text Feature Acquisition (37.7%), Backward Matching (32.0%), and the initial Mask Acquisition (29.7%). The analysis also highlights the efficiency of the neutral point processing module, which constitutes only 0.1% of the total computational cost. This low figure indicates that the boundary refinement step is achieved with minimal performance overhead.

Notably, despite the aforementioned bottlenecks, our method's runtime holds a significant advantage over mainstream methods, which typically require several hours of processing. For instance, in our evaluation on the LERF dataset, we found that InstanceGaussian (Li et al., 2025) requires approximately 140 minutes for the 3D Gaussian training phase alone. Furthermore, our model offers potential for even greater speed. In principle, it processes each category independently, allowing for significant acceleration through parallelization. However, as a key design goal is to ensure deployability on consumer-grade hardware, this imposes a constraint on the model's total memory footprint. Consequently, we did not pursue further parallelization in the current implementation.

Table 9: Component-wise runtime breakdown for our method on the `teatime` scene in the LERF dataset. The analysis highlights that VLM inference and backward matching are the primary computational bottlenecks. All timings are in seconds, measured on a single NVIDIA Tesla V100 (32GB) GPU.

| Component | Time (s) | Time / Cat. (s) | Compute % |
| --- | --- | --- | --- |
| *Computational Stages* | | | |
| Mask Acquisition | 156.99 | 1.1984 | 29.7% |
| Backward Matching | 169.23 | 1.2919 | 32.0% |
| Neutral Point Processing | 0.54 | 0.0041 | 0.1% |
| VLM Text Feature Acquisition | 199.67 | 1.5242 | 37.7% |
| CLIP Feature Extraction | 2.63 | 0.0201 | 0.5% |
| Total Computation | 529.06 | 4.0386 | 100.0% |
| *I/O Stages* | | | |
| Data Loading | 16.12 | – | – |
| Saving Output | 9.96 | – | – |
| Grand Total (incl. I/O) | 555.14 | – | – |

## D    ANALYSIS OF FAILURE CASES

**Impact of Mask Inaccuracy.** Our method demonstrates considerable robustness to sporadic segmentation errors, provided that the initial masks generated by DAM2SAM are generally accurate. However, when these masks suffer from large-scale or frequent inaccuracies, our model can produce erroneous foreground-background distinctions during the backward weight accumulation process. This, in turn, adversely affects the final segmentation accuracy, as illustrated in a failure case in Fig. 12(a).

**Mismatches from the VLM.** Incorrect matching can also arise from the VLM itself, attributable to two primary sources, as shown in Fig. 12(b). First, ambiguous segmentation masks or challenging viewing angles in the input images can provide misleading guidance to the VLM. Second, inherent limitations in the VLM's comprehension capabilities can lead to incorrect judgments even with clear inputs. Either type of error can result in incorrect category assignments, ultimately causing the point clusters to be mismatched with the intended text query.

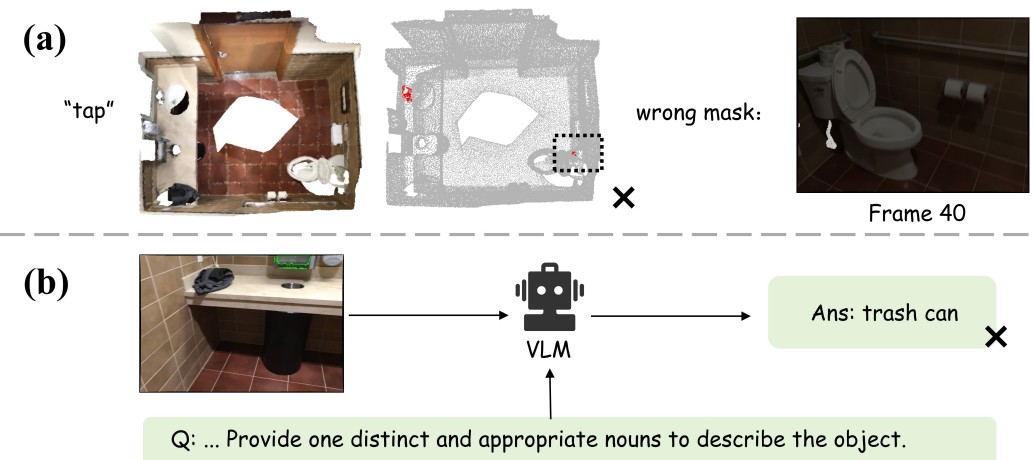

Figure 12: Examples of Failure Cases. (a) Inaccurate Masks: The segmentation model outputs incorrect 2D object masks. (b) VLM Misunderstanding: The VLM provides an incorrect object name for the given input images.

## E DISCUSSION

### E.1 DIVERSITY OF SEMANTIC CATEGORIES

Prior work has noted that a single Gaussian point can belong to multiple semantic categories (Shen et al., 2024; Qin et al., 2024; Shi et al., 2024). To verify this phenomenon, we conduct a statistical analysis of the semantic categories for all 3D Gaussian points within the `teatime` scene of the LERF dataset, as illustrated in Fig. 13. Our analysis reveals that approximately 25% of all visible 3D points exhibit multi-dimensional semantic attributes. In the context of our model, this means a substantial portion of 3D Gaussian points inherently possess multiple semantic labels simulta- neously. For instance, a single point on a tree branch may belong to the categories of "branch", "tree", and "vegetation" all at once.This phenomenon is consistent with how humans perceive 3D environments.

This semantic diversity suggests that relying on a single semantic label is often insufficient to com- prehensively describe the properties of a point. Therefore, this inherent polysemy must be fully considered when performing 3D semantic understanding.

### E.2 NEUTRAL POINTS

Prior work on so-called "boundary points" (Li et al., 2024; Zhang et al., 2025) has primarily focused on refining their positions through dedicated training strategies to enhance semantic understanding. However, while repositioning these boundary points can improve semantic segmentation accuracy, it often compromises the realism and fidelity of the final rendering. This trade-off arises because boundary points include a special subset of points that belong neither to the foreground nor the background. These points serve as transitional elements that are crucial for ensuring rendering realism but lack specific semantic meaning. We term these as **neutral points**.

Neutral points are abundant in 3DGS scenes, making them non-negligible for semantic understand- ing. Nevertheless, accurately identifying and removing these neutral points in an unsupervised man- ner remains a significant challenge. In our implementation, precisely filtering out these points during the matching stage is difficult due to computational efficiency constraints. In Fig. 14, we present the visualization of neutral points from our model on the LERF dataset. Developing more effective methods to model and eliminate neutral points is a key direction for future improvement of our method.

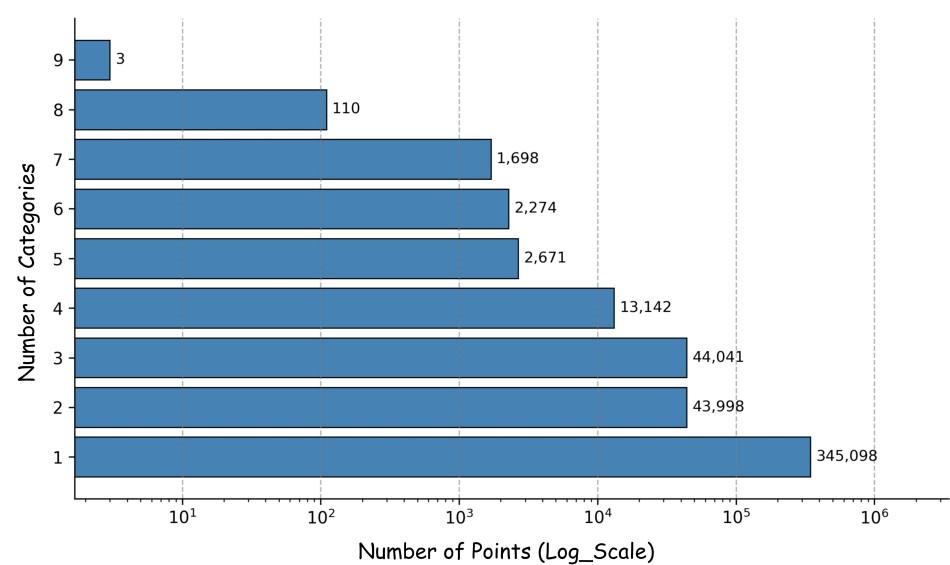

Figure 13: Category distribution of visible Gaussian points in the `teatime` scene from the LERF dataset.

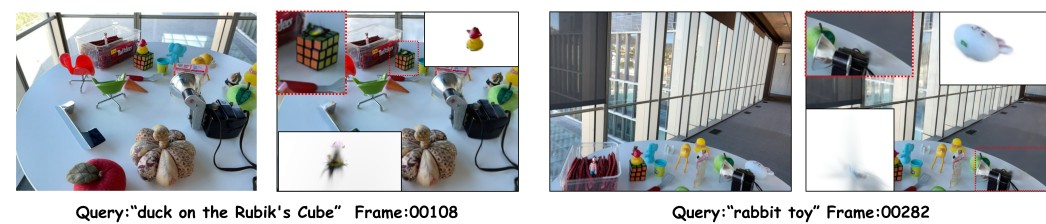

Query:"duck on the Rubik's Cube"  Frame:00108        Query:"rabbit toy"  Frame:00282

Figure 14: Qualitative results for rendering foreground, neutral, and background points on the `figurines` scene from the LERF dataset.

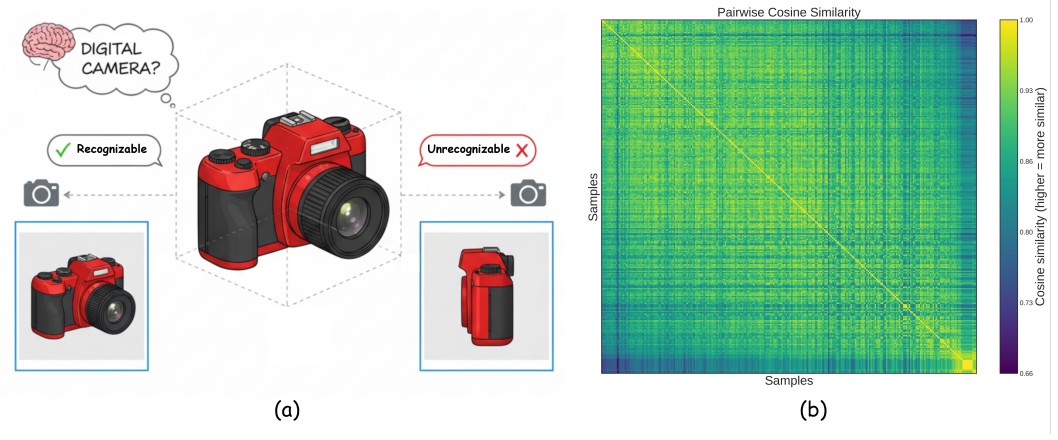

(a)                                             (b)

Figure 15: Illustration of the inconsistency of semantic features across different viewpoints. (a) The same object can present different semantic characteristics from different viewpoints. (b) Visualization of feature similarity for the "Jake the Dog" object in the `figurines` scene. The plot shows the cosine similarity scores between feature vectors from different views; a higher value (closer to 1) indicates that the features are more similar.

### E.3 INCONSISTENCY OF SEMANTIC FEATURES

Our work diverges from the common practice in related literature of feeding masked object regions into a CLIP image encoder to obtain semantic features. This decision is based on the observation that for the same object, its semantic features can exhibit significant variations across different viewpoints (Cen et al., 2025). As shown in Fig. 15(a), acquiring accurate CLIP image features becomes more challenging from certain angles. Due to the existence of such views, strategies like selecting the features from the view with the largest mask area or averaging the features across all views inevitably introduce errors.

To validate this phenomenon, we selected the "Jake the Dog" object from the `figurines` scene in the LERF dataset and extracted its CLIP image features from multiple viewpoints. A visualization of these features is presented in Fig. 15(b). The figure clearly shows that even for the same object, the semantic features vary noticeably with the observation angle. This feature inconsistency suggests that conventional strategies based on single-mask or averaged-mask feature extraction can lead to information loss, thereby degrading matching performance. In contrast, our VLM-based feature extraction approach alleviates this issue to a certain extent, enhancing the stability and robustness of the semantic representation.

