# OpenReview forum: "Polysemous Language Gaussian Splatting via Matching-based Mask Lifting"
_ICLR.cc/2026/Conference — ICLR 2026 Conference Withdrawn Submission_

### Official Review · Reviewer_5tEV · 2025-10-25

**Soundness:** 3
**Presentation:** 3
**Contribution:** 2
**Rating:** 4
**Confidence:** 5

**Summary:**

The paper proposes MUSplat, a training-free pipeline for open-vocabulary understanding on 3D Gaussian Splatting (3DGS) scenes. Instead of per-scene semantic optimization, it: (1) lifts multi-granularity 2D masks into 3D via back-projection, (2) refines boundaries by removing “neutral” points using semantic entropy + opacity filtering, and (3) derives instance-level textual features by asking a VLM to name the object from a few key views and encoding those names with CLIP for matching.

**Strengths:**

1. This paper is well-written and has a clear problem framing and motivation. And the method to use VLM working as an auto-vocabulary annotator is quite interesting.
2. The Neutral Point Processing part is tackling the problem of "how should we deal with the gaussians that in the edge of 2 objects". This is the first paper I saw that explicitly try to solve this problem.
3. Strong empirical results with compelling efficiency. LERF object selection: MUSplat outperforms previous best by +10.7 mIoU on average (Tab. 2).  ScanNet OV segmentation: consistent SOTA across 19/15/10-class settings (Tab. 3).  Storage & memory: CLIP feature storage drops from ~3 GB → ~3 MB; peak VRAM ~8 GB (Tab. 1).  End-to-end runtime on a complex LERF scene ≈ 9.25 min with a breakdown that identifies VLM inference and backward matching as the main costs (Tab. 9).
4. The paper did a pretty comprehensive ablation study on every module they designed in the paper.

**Weaknesses:**

1. There are some similar works not mentioned in the paper. The paper is comparing mostly with the optimization based method. However, some optimization-free methods are not compared. I would highly think about raise the score if you can add the comparison with optimization-free method. e.g. [1] OccamLGS, [2] LUDVIG [3] Gradient-Weighted 3DGS. In addition, comparison with generalizable method is also encouraged, e.g.  [4] SceneSplat [5] SceneSplat++

[1] [BMVC 2025] Occam’s lgs: A simple approach for language gaussian splatting.
[2] [ICCV 2025] LUDVIG: Learning-Free Uplifting of 2D Visual Features to Gaussian Splatting Scenes
[3] Gradient-Weighted Feature Back-Projection: A Fast Alternative to Feature Distillation in 3D Gaussian Splatting
[4] [ICCV 2025] SceneSplat: Gaussian Splatting-based Scene Understanding with Vision-Language Pretraining
[5] [NeurIPS 2025] SceneSplat++: A Large Dataset and Comprehensive Benchmark for Language Gaussian Splatting

2. While the paper argues polysemy and also have this word in the title, and provides a distributional discussion, the evaluation still uses standard single-label queries. More explicit multi-label / attribute-rich benchmarks (e.g., “wooden chair with cushion”) would better validate the claimed advantage. For example, run the scannet200 and compare on this might be an option. Authors are encouraged to think more on how to demonstrate this.

**Questions:**

1. The more comparison with optimization-free methods would be good.
2. Can you test smaller open-sourced VLMs to see the performance?
3. The time in Tab 1 seems to not include the preporocessing time of SAM + CLIP and also VLM. Can you specify the training time including all of the preprocessing? And I assume that the outputed language 3DGS size is almost similar with LangSplat?

---

### Official Review · Reviewer_evWQ · 2025-10-29

**Soundness:** 2
**Presentation:** 1
**Contribution:** 1
**Rating:** 2
**Confidence:** 4

**Summary:**

The method proposes an algorithm for the 3D open vocabulary semantic segmentation method. This submission propose object-level grouping (Section 3.2) and instance feature extraction (Section 3.3). These two modules are to lift 2D understanding on top of 3D Gaussians.

The object-level grouping in Section 3.2 is to life category-agnostic 2D masks into 3D Gaussians. The process itself is the same as Gaussian grouping task [D]. While the performance itself outperform [D], I cannot find the high-level difference between [D] and the proposed method. Moreover, it is quite difficult to understand the description about 'neural point processing'.

The instance feature extraction in Section 3.3 is to assign text embedding on top of 3D Gaussians. The captions or tags are extracted from the existing methods, DAM2SAM and Gemini.

__Related works__
[D] Gaussian grouping, ECCV 24

**Strengths:**

Overall, the performance is good compared to recent studies. It is highly promising that the evaluation is conducted in LeRF dataset as well as ScanNet dataset. Honestly, except this, I cannot fine any strengths in this submission.

**Weaknesses:**

__W1. Wrong statements__

_W1-1. (line 82) "the prevailing contrastive methods are restricted by a one-to-one semantic assignment, attributing a single semantic concept to each Gaussian."_
Concretely speaking, previous papers learns and encode language 'features' on top of 3D Gaussians. While each feature is mapped to the text prompt, such as 'chair' or 'table', via text embedding from CLIP text encoder, the idea of contrastive learning is to learn 'relativeness'. In other words, even though an encoded feature vector on one Gaussian is not exactly same as the text embedding from 'chair', the Gaussian can be classified or segmented as 'chair' if the encoded feature vector is 'relatively' close to the text embedding from 'chair'. This understanding is derived from the definition of the contrastive loss.

For more clearance, let me provide one more example. Given encoded Gaussian features from previous works, we can assign different category on each Gaussian in the ScanNet20 benchmark and ScanNet200 benchmark. This supports that 'one-to-one semantic assignment' stated in this submission is wrong. Please refer to more details in the pointcloud-based 3D open vocabulary segmentation methods [A,B,C].

__W2. Wrong understanding of previous works.__

_W2-1 (line 50) "and subsequently assigns these compressed features to the 3D Gaussians"_
This is wrong. Dr.Splat optimizes codebook using Product Quantiazation first, and then assign the indices of PQ table on top of each 3D Gaussian that maps to CLIP embeddings.

_W2-2 (line 148) "Dr.Splat trains a feature compressor for each scene"_
The feature compressor in Dr.Splat is the Product Quantization algorithm. The resulting codebook, named PQ table, is trained by LVIS dataset. Once it trained, it does not require per-scene optimization. Accordingly, the statement in this submission is wrong.

__W3. Weak novelty and motivation__
Overall, the performance improves a lot compared to the recent studies. However, I do not feel that the proposed modules, Sections 3.1, 3.2, and 3.3, are novel. Moreover, it is difficult to tell which modules are particularly effective for the performance improvement. While the authors proposed ablation studies, it is not clearly resolved my concerns.

Moreover, the pipeline consists of (1) Gaussian grouping (section 3.2) and (2) text assignment per grouped Gaussian. I read the introduction section but cannot conclude myself why the authors propose such a design. There is no link between the addressed problems / motivation and the proposed solutions. For myself, I do emphasize this aspect more and the state-of-the-art performance is not the top priority. Accordingly, I cannot say this submission is novel.

__W4. Limited ablation study__
I addressed this problem in several aspects through the review. One additional example is that the Instance feature extraction is not something new. There are previous papers that utilize the 2D vision language models to extract point-text paired data. For example in [A], this paper also provides ablation study by using different 2D vision language models and the caption generation pipelines. In this perspective, the authors should have provided ablations studies to support the validity of the pipeline in Section 3.3.

_W4-1. missing details_
One additional comment is that I wonder how the authors deal with merging the captions from multi-view masks. Based on the description in Section 3.3, I cannot catch how it proceeds with.

__Related works__
[A] Mosaic3D, CVPR 2025
[B] RegionPLC, CVPR 2024
[C] OV3D, CVPR 2024
[D] Gaussian grouping, ECCV 24
[E] Gaga: Group Any Gaussians via 3D-aware Memory Bank, arXiv 2024
[F] Liang et al. Mind the gap, NeurIPS 2022
[G] Mistretta et al. Cross the gap, ICLR 2025.
[H] OpenScene, CVPR 2023

**Questions:**

__Q1. Minor__

_Q1-1. (line 40) "the efficiency of neural techniques"_
In this statement, the authors summarize the strength of 3DGS as above. But I wonder what is the meaning of 'neural techniques'?

Unlike NeRF, 3DGS proposes 3D Gaussian representation with tile-based rasterization and splatting algorithm. The authors also mention the first difference, 3D Gaussian representation is the explicit 3D representation. However, I do not agree that rasterization and splatting algorithms are 'neural techniques'.

Can authors elaborate the terminology, 'neural techniques'?

__Q2. Object-level grouping__
The proposed method in Section 3.2 of the manuscript is related to Gaussian grouping [D,E]. Can authors elaborate the technical differences between the proposed methods and [D]? In a minor detail, there are some difference, but it looks highly similar in my understanding.

__Q3. What is the dominant contributions for the performance improvement?__
Unlike paper flow, I personally suspect that the dominant performance improvement is derived from Section 3.3 Instance feature extraction. As widely known, there is a modality gap in CLIP vision embeddings and CLIP text embeddings [F]. Moreover, another recent study [G] shows that text-to-text search is more effective in CLIP embedding. Similarly in pointcloud-based 3D open vocabulary methods, early work [H] use CLIP vision embedding while the recent studies use CLIP text embedding [A,B,C]. These analysis and tendency show that using text embedding is effective for the 3D open-vocabulary segmentation task.

_Q3-1. Based on this observation, I cannot tell whether the other contributions are truly effective for the performance improvement. There are no ablation studies at all._

_Q3-2. Furthermore, I cannot clearly tell the novelty in the Section 3.3_
I believe that the proposed technique can be also applicable to pointcloud, which is not specialized in 3D Gaussians. Moreover, the proposed method in Section 3.3 is highly similar to the data generation pipeline by Mosaic3D [A]. Can authors resolve my concern regarding this issue?

__Q4. Training-free?__
Can authors explain why the proposed method is training-free? Even in the Table1, the training time for the proposed method is __None__. Given the proposed method in Section 3.2 and Section 3.3, it still requires some more time to be processed for the inference. In my understanding, for the object-level grouping, it requires some parameters to be optimized. Can authors clarify the definition of training time in the Table 1?

__Related works__
[A] Mosaic3D, CVPR 2025
[B] RegionPLC, CVPR 2024
[C] OV3D, CVPR 2024
[D] Gaussian grouping, ECCV 24
[E] Gaga: Group Any Gaussians via 3D-aware Memory Bank, arXiv 2024
[F] Liang et al. Mind the gap, NeurIPS 2022
[G] Mistretta et al. Cross the gap, ICLR 2025.
[H] OpenScene, CVPR 2023

**Details Of Ethics Concerns:**

No needs any ethics review.

---

### Official Review · Reviewer_VqC3 · 2025-10-30

**Soundness:** 3
**Presentation:** 3
**Contribution:** 2
**Rating:** 4
**Confidence:** 2

**Summary:**

The paper introduces a method for language embedded 3DGS. Authors observe that prior approaches inherit three limitations: (1) reliance on costly per-scene optimization, (2) design that encodes only simple semantics, and (3) multi-view inconsistency. To resolve these, the authors propose a training-free pipeline, textual features that capture multi-level semantics, and a matching-based integration. The core idea is 3D grouping and matching for semantic feature updates: track to obtain multi-view-consistent instance masks, group the corresponding 3D instances, then extract VLM textual features from the view where the instance is viewed largest and assign them as a dictionary to enforce view consistency. Various Experiments show proposed method works in training-free approach with lower memory across several benchmarks.

**Strengths:**

- Broad experimental and application demonstrations.
- Strong motivation: tackles multi-view inconsistency and training dependence, which are important issues recognized by the community.

**Weaknesses:**

**Table 2 comparison (pixel- vs. point-based)**
- The pixel-based setting optimizes through a feature rendering loss after rasterization, which appears to pursue a different goal from the proposed training-free, 3D-activation approach. This comparison seems unnecessary—especially since the “-m” models are already modified versions.

**Polysemy is not directly evidenced**
- Although text based embeddings are used to capture polysemous semantics, there is no explicit example showing robust multi-concept activation for the same Gaussian. Instead of the indirect discussion or explanation in Discussion E.1, a direct experiment contrasting this with CLIP-based baselines would be more convincing. (case where CLIP based method fails but multi-dictionary based method succeed)

**Minor (but important) clarifications around explanation on previous work**
- Line 51: It seems not “compress then uplift,” but aggregate then compress.
- Lines 148, 78: If there is one-time pre-training kept reused later, so for each scene, the computation overhead is unnecessary.
- This also raises concern with Table 1 training time: the cited Dr. Splat paper reports much shorter uplifting time than OpenGaussian, yet your table shows similar times. Please reconcile these discrepancies; otherwise it might invites doubt about other experimental results.

**Questions:**

All questions are covered in the Weaknesses above.

---

### Official Review · Reviewer_Z9Nr · 2025-10-31

**Soundness:** 3
**Presentation:** 3
**Contribution:** 2
**Rating:** 4
**Confidence:** 3

**Summary:**

This paper presents MUSplat, a training-free framework for open-vocabulary understanding in 3DGS. The method operates by first lifting multi-view, multi-granularity 2D masks into 3D to form initial object groups via a back-projection technique. It then refines the boundaries of these groups by identifying and filtering out ambiguous "neutral points" based on their semantic entropy and geometric opacity. To enable open-vocabulary querying, the approach distills the visual appearance of objects from their key viewpoints into robust textual features using a Vision-Language Model (VLM), thereby aligning the 3D groups with language queries. The reported results demonstrate that MUSplat achieves competitive performance on open-vocabulary tasks while eliminating the need for per-scene training, offering a computationally efficient alternative to existing paradigms.

**Strengths:**

1.  The paper is well-written and clearly structured, making the proposed methodology easy to follow.
2.  The experimental evaluation is comprehensive, demonstrating impressive results on mainstream benchmarks. Furthermore, the ablation studies convincingly validate the design choices of the framework.
3.  The core idea of a training-free framework is novel and interesting, presenting a distinct paradigm for semantic understanding in 3DGS.

**Weaknesses:**

1. The paper positions its framework as a offline post-processing pipeline, which fundamentally diverges from the core philosophy of 3DGS as an explicit and editable scene representation. Mainstream approaches (e.g., LangSplat, LEGaussian) strive to internalize semantic attributes as inherent features of the Gaussians, or (e.g., GS-Grouping, OpenGaussian) learn their structured semantic relationships, with the goal of constructing a persistent and real-time queryable semantic field. In contrast, the offline design of this work essentially treats semantics as temporary, external query results rather than intrinsic properties that the 3DGS scene should inherently possess. This undermines the value of the proposed method in dynamic and interactive applications and reflects a conservative and limited perspective in its task formulation.

2. The paper only reports the total training time of comparative methods. However, it should be noted that the training process in these methods typically involves the joint optimization of both geometric and semantic properties. In contrast, the proposed method inherently assumes the pre-existence of well-trained Gaussian geometries. This discrepancy raises concerns regarding the fairness of the presented computational comparisons.

3. The performance of the proposed model is heavily reliant on several third-party models and involves a relatively complex processing pipeline. Furthermore, while many existing works achieve real-time rendering capabilities during inference, the method described in this paper depends heavily on extensive post-processing steps. Notably, the paper does not provide a comparative analysis of inference times, which is a critical aspect for evaluating practical utility.

4. The performance advantage demonstrated in the experiments seems to stem primarily from two sources: the use of a more advanced VLM backbone and the adoption of a post-processing paradigm that avoids the difficult problem of intrinsic semantic modeling in 3DGS. While effective, this raises a question about the long-term viability of this approach. Does the proposed framework provide a foundational step towards building persistent, editable semantic scenes, or does it primarily highlight the performance ceiling achievable by bypassing the core challenge through extensive post-processing?

**Questions:**

See the description in the "Weakness" section.
In addition, I would like to ask another question: whether the input views in this paper, like those in other models, consist of a fixed sequence of views under a specific scenario, or if there are additional rendered perspectives involved?

---

### Note · Authors · 2025-11-13

I have read and agree with the venue's withdrawal policy on behalf of myself and my co-authors.